# Understanding a Mechanistic Basis of ABA Involvement in Plant Adaptation to Soil Flooding: The Current Standing

**DOI:** 10.3390/plants10101982

**Published:** 2021-09-22

**Authors:** Yancui Zhao, Wenying Zhang, Salah Fatouh Abou-Elwafa, Sergey Shabala, Le Xu

**Affiliations:** 1Hubei Collaborative Innovation Centre for Grain Industry, Engineering Research Centre of Ecology and Agricultural Use of Wetland, Ministry of Education, Yangtze University, Jingzhou 434025, China; ZhaoYancui_zyc123@163.com (Y.Z.); wyzhang@yangtzeu.edu.cn (W.Z.); 2Agronomy Department, Faculty of Agriculture, Assiut University, Assiut 71526, Egypt; elwafa75@aun.edu.eg; 3Tasmanian Institute of Agriculture, University of Tasmania, Private Bag 54, Hobart 7001, Australia; 4International Research Centre for Environmental Membrane Biology, Foshan University, Foshan 528000, China

**Keywords:** abscisic acid, phytohormones, morphological adaptations, flooding stress

## Abstract

Soil flooding severely impairs agricultural crop production. Plants can cope with flooding conditions by embracing an orchestrated set of morphological adaptations and physiological adjustments that are regulated by the elaborated hormonal signaling network. The most prominent of these hormones is ethylene, which has been firmly established as a critical signal in flooding tolerance. ABA (abscisic acid) is also known as a “stress hormone” that modulates various responses to abiotic stresses; however, its role in flooding tolerance remains much less established. Here, we discuss the progress made in the elucidation of morphological adaptations regulated by ABA and its crosstalk with other phytohormones under flooding conditions in model plants and agriculturally important crops.

## 1. Introduction

Flooding is a major environmental constraint affecting crop production systems. While it is particularly acute in Asia [1], this is a world-wide problem that will come with a significant economic cost [2]. In light of climate change and global warming, it is anticipated that the future threats of flooding are likely to increase, affecting more regions [3]. Thus, the importance of dealing with impacts of flooding on the agricultural sector has emerged as one of the critical issues for food security [4]. The term flooding encompasses both waterlogging and submergence conditions. Waterlogging refers to the conditions in which the root-zones are saturated with excessive water, whereas submergence is defined as a situation in which the aboveground parts of plants are partially or completely covered by water [5]. The adversity of flooding stress is largely due to the dramatically restricted availability of oxygen for submerged plant tissues since gases’ diffusion is many-fold slower in water than in air [6]. Reduced O_2_ diffusion in saturated soils suppresses aerobic respiration and subsequently causes an energy crisis and accumulation of toxic substances [7]. Waterlogging of the rhizosphere or partial flooding of the aboveground parts of a plant lead to a gradual hypoxia (deficiency of oxygen), and long-term complete submergence brings about anoxia (the total absence of oxygen [8]).

Plants have developed several strategies to cope with the adverse consequences of submergence conditions. One strategy is known as the quiescence strategy where plants do not elongate shoots under flooding conditions, to minimize energy and carbohydrate consumption, but continue to regrow after stress. This strategy has been efficiently implemented by wetland species, which could survive relatively deep and transient floods (e.g., *Rumex acetosa*) [9,10,11,12,13]. Another strategy (known as an escape strategy) is utilized by plants such as deepwater rice and the wetland dicot *Rumex palustris*. Here, plants rapidly extend their petioles or stems to allow leaves to reach the water surface to aerate the remainder of the plant [10,14,15,16,17]. The escape strategy is effective under prolonged, but relatively shallow, flooding events [17,18]. 

The key regulator of these flooding-induced acclimations is ethylene (ET), which, owing to its gaseous nature, rapidly accumulates in flooded tissues [6,14,19]. In addition, a permeability barrier for ET in the roots prevents ET losses and leads to the accumulation of high levels of ET in root tissues [20]. ET is an early and reliable signal for different complementary pathways required for morphological and anatomical acclimations under flooding conditions, like aerenchyma formation, adventitious roots (ARs) formation, and shoot growth of plants in response to flooding stress. The rapid accumulation of ET may work as a priming factor for these adaptive responses of plants to flooding [8]. 

A wide range of studies found that interactions of ET with other phytohormones, and specifically, ABA, were also important for adaptations to flooding stress. ABA is known as a “stress hormone” that accumulates under different abiotic stresses and mediates cross-adaptation in plants [21]. Under drought, salinity, or cold stress, ABA accumulation causes stomatal closure to conserve water in leaves [22,23,24]. However, compared with ET or GA (gibberellin), the role of ABA in flooding tolerance remains much less established. The aim of this work was to fill this gap in the knowledge and review the progress made in the elucidation of morphological adaptations regulated by ABA and its crosstalk with other phytohormones under flooding conditions. 

## 2. Changes in ABA Content and ABA-Regulated Responses during Submergence 

The changes of ABA content during flooding stress are summarized in Table 1, and they are species-dependent. In an early study, ABA was considered as a potent antagonist of GA action in rice internodes through influencing GA responsiveness, evidenced by both applied ethylene and submergence, which reduced endogenous ABA levels in internodes significantly within 3 h [25]. The submergence- and ethylene-mediated decline of ABA levels in deepwater and lowland rice has also been demonstrated by researchers [26,27,28]. 

The accumulated ET mimics a submerged environment and reduces ABA concentration via activating the ABA breakdown to inactive ABA catabolite phaseic acid in rice [29]. In addition, under completely submerged conditions, the accumulated ET in *Rumex palustris* causes a massive depletion of ABA levels, and the decline in the endogenous ABA concentration stimulates the expression of *gibberellin 3-oxidase* (*GA3OX1*), which catalyzes the conversion to bioactive gibberellin GA1 [11,12]. 

However, ET does not significantly alter the expression of genes involved in ABA metabolism, as is evident from the analysis of several *Arabidopsis* ET receptor mutants [30]. In another study, the *Ethylene Response Factor VII* (*ERF-VII*) transcription factors *RELATED TO APETALA2.12* (*RAP2.12*), *RAP2.2*, and *RAP2.3* were all induced by exogenous ABA treatment in *Arabidopsis*. In addition, the enhanced ABA sensitivity of the *RAP2* genes overexpressors might be due to the activation of abscisic acid insensitive 5 (*ABI5*) [31]. Moreover, hypoxia disturbs ABA metabolism and increases ABA sensitivity in embryos of dormant barley grains [32]. Similarly, two perennial ryegrass accessions exhibited a dramatic decrease in ABA content, which resulted from enhanced ABA catabolism via the upregulation of *ABA8Ox1*, although the expression of zeaxanthin epoxidase and 9-cis-epoxycarotenoid dioxygenase 1 (*NCED1*) was upregulated under submergence stress [5]. The decrease of the ABA content in plants during flooding stress was also reported in *Solanum dulcamara*, *Scirpus mucronatus*, and *Carrizo citrange* [33,34,35].

**Table 1 plants-10-01982-t001:** Changes of ABA concentration affected by flooding stress in different plant species.

Stress	Species	Tissue	Response	Reference
flooding	*Solanum dulcamara*	stems, AR primordia	decreased	[11,33]
flooding	*Rumex palustris*	petioles	sharply decreased	[36]
submergence	*Lolium perenne*	leaf	decreased	[5]
submergence	*Oryza sativa, deepwater*	internodes	decreased	[25]
submergence	*Oryza sativa, lowland*	shoot	decreased	[28]
submergence	*Scirpus micronatus*	shoots	decreased	[34]
flooding	*citrus*	root	decrease	[37]
submergence	*Rumex palustris*	petioles	decrease significantly	[11]
submergence	*Nasturtium officinale*	petioles, stem	decrease	[38]
waterlogging	*Carrizo citrange*	roots	decreased, back to control	[35]
flooding	*citrus*	leaf	sharp increase	[37]
flooding	*Pilsum sativum* L.	shoot	increase	[39]
flooding	*Pisum sativum* L.	root	early increase	[39]
flooding	*Nicotiana tabacum* L.	leaf	increase	[40]
waterlogging	*Vigna radiata* L.	leaf	increase	[41]
anoxia	*Lactuca sativa* L.	roots	unchanged	[42]
flooding	*Glycine max*	seedlings	unchanged	[43]

However, in other species, such as *Nasturtium officinale* (watercress), ABA breakdown is not likely regulated by ET, since ACC (l-aminocyclopropane-l-carboxylic acid) application did not alter *NoNCED3* or *NoCYP707A1/A2* gene expression, and submergence did not induce GA biosynthesis genes in this species [38]. Taken together, this suggests that although ABA depletion is a common feature in hypoxic plant tissues, it might be independently regulated in a species-specific manner rather than necessarily being under ET control. 

On the contrary, elevated levels of ABA content in response to flooding have also been reported in the leaves of alfalfa, pea, and tobacco, as well as in the roots of *Gerbera jamesonii* [40,44,45,46]. Meanwhile, the endogenous ABA concentration remained unchanged in response to flooding in soybean (*Glycine max*) and lettuce (*Lactuca sativa*) [42,43]. 

The above controversy in results poses a question as to the real relationship among ABA, ET, and submergence-induced reactions. It could be suggested that plants grow quickly to escape submergence stress by the fast drop in the endogenous ABA level, while hypoxia-susceptible plants usually respond to oxygen shortage by a drastic increase of the ABA concentration [8]. 

Another routine to study the function of phytohormones is to measure the response of plants after the exogenous phytohormone application of mutants (see Table 2 for selected examples). 

The interaction between flooding and other soil factors, such fertilizers and pollutants, needs to be accounted for, as ABA levels in plants are subjected to orchestrated actions exerted by these factors. For example, the concentration of ABA in the soil solution was highest in acid soils and in soils with reduced moisture, while it was lowest in moist, neutral, and moderately alkaline soils [47]. Pollutants, like heavy metals and other toxic substances, result in a shift in the biological activity of the soil [48,49]. ABA content is also much higher in plants grown under conditions of soil salinity [50] and changes in plant ontogeny [51]. 

The climate also exerts a major impact on ABA production; thus, plant responses to flooding in a dry/warm climate will be strikingly different from those under humid or cold conditions. For example, ABA content is significantly higher in cold-grown plants (compared with optimal temperature conditions) [52]. This difference in the basal ABA levels will impact patterns of ABA signaling and, hence, plant adaptation to hypoxia. 

**Table 2 plants-10-01982-t002:** The effect of ABA on flooding stress in different plant species.

Species	Chemicals	Concentrations (uM)	Effect	Treatment	Reference
*Lactuca sative* L.	ABA	1, 3, 10, 30, 100, 300	increase survivability	anoxia 24 h	[42]
*Glycine max*	ABA	5, 10, 50	increase the survival	flooding	[43]
*Oryza sativa*	ABA	0.1 uM, 24 h	improving seeds resistance	submergence	[53]
*Arabidopsis*	ABA	10, 50, 100	increase tolerance	anoxia	[54]
*Zea mays.* L.	ABA	100 uM, 24 h	increase tolerance	anoxia	[55,56]
*Nasturtium officinale*	ABA	0.5, 1, 5	inhibits shoot elongation	submergence	[38]
*Rumex palustris*	ABA	-	inhibits petiole elongation	submergence	[57]
*Oryza sativa*	ABA	1	inhibits petiole elongation	submergence	[25]
*Oryza sativa*	fluridone (biosynthesis inhibitor)	-	induced AR emergence	submergence	[58,59,60]

## 3. Stomatal Closure and Root-Shoot Response 

Stress-induced ABA-mediated stomata closure has been observed in plants exposed to hyperosmotic conditions, such as drought or salinity [22,23,24]. Surprisingly, plants also close stomata in response to waterlogging. This process delays leaf chlorosis and senescence, although the underlying mechanisms are poorly understood [61].

ABA is known to act in root-to-shoot signaling in the drought response; it is therefore reasonable to suggest that ABA may act as a mobile signal for root-to-shoot communication during waterlogging stress as well [62]. However, a transcriptome study performed on *Arabidopsis* revealed an upregulation of ABA biosynthesis genes in leaves and a downregulation in roots under flooding conditions [63], which was reflected in the actual ABA levels in shoots and roots. Interestingly, *NCED3* was slightly induced in the ET insensitive mutant *ein2-5*, indicating that ET alters ABA biosynthesis and signaling in systemic responses [63]. The ABA concentration was also decreased in the xylem sap in flooded tomato roots [64]. It was suggested that ABA may accumulate in leaves of flooded plants because of reduced translocation of photoassimilates out of leaves, and roots do not act as the source of ABA because most roots collapse rapidly and die within the first few days of flooding [45]. This model was further supported by experiments with wilty mutants of peas and tomatoes, showing that ABA accumulation in the leaves of flooded plants is determined by the shoot rather than the root genotype [45]. This notion is rather similar to the modern view of ABA-mediated signaling in response to salinity and drought [22,23,24] that also questions the root origin of ABA signals. 

The nature of the mobile signal that mediates stomatal closure remains elusive, and no solid evidence could support the transport of phytohormones from the root to the shoot prior to flooding-induced stomatal closure [65,66,67]. In waterlogged citrus leaves, JA (jasmonic acid) levels rapidly but transiently increased preceding the progressive accumulation of ABA. In citrus roots, as indole-3-acetic acid increased, JA and ABA levels decreased rapidly and significantly in all genotypes under flooded conditions [37]. Similarly, the ET precursor ACC, JA, or metabolite fluxes might contribute to systemic flooding stress adaptation. 

The morphological adaptations regulated by ET, GA, and ABA are summarized in Figure 1. 

## 4. Petioles Elongation and Hyponastic Growth 

For plants using the escape strategy, the accumulation of ET causes a reduction of ABA content, which subsequently enhances the concentration of bioactive GA and plants’ sensitivity to GA [30,75]. GA is the ultimate hormone that stimulates both cell division in the intercalary meristem and internodal elongation. Under hypoxic conditions, the exogenous application of GA upregulates the activity of ACC synthases (*OsACS5*) in both lowland and deep-water rice seedlings, whereas the application of ABA has an adverse effect on the activity of *OsACS5* [76]. The external application of ABA to deep-water rice and *Rumex palustris* restricts submergence-induced petiole elongation, indicating that ABA may function as a negative regulator of enhanced petiole growth [11,76]. The adverse effect of ABA on petiole growth could be reversed by GA, suggesting that submergence-induced growth is manipulated through the balance between ABA and GA [30]. Similarly, the rapid petiole growth of lotuses might also result from an altered balance between growth-inhibiting ABA and growth-promoting GA hormones [77], summarized in Figure 1. 

The proposed sequence of events mediated by ET, GA, and ABA is that a low oxygen concentration leads to ET accumulation, which causes a decrease in ABA levels and an increase in GA1 concentration as well as GA sensitivity, and internodal elongation was promoted ultimately [78,79]. Different endogenous ABA contents determine different petioles elongation rates of two accessions of the wetland plant *Rumex palustris* under submerged conditions [80]. In addition, an elongating rice variety exhibited a stronger reduction in ABA concentration than a non-elongating variety [14]. ABA could inhibit H^+^ pumps in guard cells, and the reduced ABA content might stimulate acidification of the apoplast [18]. In addition, auxin can facilitate the activation of plasma membrane H^+^-ATPases, which leads to apoplast acidification and expansin activation [18,81,82]. Secondly, ABA might work as an antagonist of GA and suppress root growth for survival during flooding. A reduction in ABA content might lead to an increased GA signal strength, which enhances starch breakdown in the shoot to generate sugars to fuel proton pump activity and provide building blocks for cell wall synthesis (Figure 1). Together with cell wall acidification, the higher expression level of expansins promoted by GA can enhance cell wall extensibility, which is beneficial for elongation. GA biosynthesis inhibitor PAC or GA3 application did not significantly influence the elongation of watercress stem or petioles, which indicates that the role that GA played in watercress shoot growth underwater might be different from other plants [38]. 

IAA also plays a role in plants’ shoot elongation under flooding stress. However, neither removal of the lamina (putative auxin source) nor addition of N-1-naphthylphthalamic acid (NPA), which inhibit the polar auxin transport, had any effect on the submergence-induced elongation of *R*. *palustris* petioles over a two-day treatment [83]. In the same work, the application of a polar IAA transport inhibitor on *Ranunculus scleratus* prevented petiole extension, and ET was found to induce polar auxin transport [83]. Interestingly, a role of auxin in *R*. *palustris* has also been identified. Inhibiting polar auxin transport, chemically or by lamina removal, transiently reduced the underwater elongation response [84]. Furthermore, a rapid accumulation of auxin in the ab- and adaxial fragments of the elongating petioles was observed [75]. In addition, internode elongation in peas requires an interaction between auxin and GA [85]. Decapitation (removal of the auxin source) results in a downregulation of GAβ-hydroxylase, responsible for bioactive GA1 synthesis, and a reduction of the GA1 concentration in pea internodes, and this down-regulation can be completely rescued by the application of IAA [85]. This is indicative of internode elongation; a certain amount of endogenous IAA biosynthesis is required by GA1. 

Hyponastic growth, together with petiole elongation, enables plants to reach the air and restore gaseous exchange [75,86,87]. The orientation of plant petioles and leaf blades changes from horizontal to almost vertical during complete submergence, and this phenomenon is known as a hyponastic growth [88]. During early submergence, a sharp decline of ABA, mediated by ET, is a prerequisite to avoid the inhibitory effect of ABA on hyponastic growth. ET-induced hyponastic leaf and petiole movement is dependent on GA [89]. In *R*. *palustris*, the lateral redistribution of auxin to the outer cell layers in the petiole is promoted by ET, and it likely leads to differential growth, given the expansion of specific cells [75]. ET was unable to induce differential cell expansion and leaf hyponasty in the *Arabidopsis* ROTUNDIFOLIA3 mutant, defective in the P450 cytochrome involved in brassinosteroids (BRs) biosynthesis. Chemical perturbation of BR biosynthesis also generates similar results, predicating the involvement of BRs in ET-regulated hyponasty [90] (Figure 1). 

## 5. Heterophylly Initiation 

Heterophylly is defined as abrupt changes in leaf morphology in a single plant in response to ambient environmental cues [91]. Leaves of constitutively submerged plants exhibit a distinct morphology with a narrow shape, lack of stomata, and reduced vessel development compared to terrestrial leaves. ET seems to be a key regulator of heterophyte formation in aquatic plants [92]. Exogenous ET application promotes the formation of submerged leaves, and the endogenous levels of ET are elevated in submerged plants compared to terrestrial plants [92,93]. Anatomical and developmental studies revealed that alterations in cell division patterns, promoted by ET, resulted in changes in leaf form. The modified cell division patterns were attributed to the overactivation of genetic networks composed of the ET signaling transducer *ET INSENSITIVE3* and abaxial genes that repress genes underlying xylem and stomatal development, while the higher levels of ABA produced in terrestrial leaves played a positive role [94,95,96]. 

Submerged leaves produced higher levels of ET but lower levels of ABA compared with terrestrial leaves [96]. The exogenous application of ABA to submerged plants resulted in terrestrial-type leaves’ formation under submerged conditions [93,97]. Moreover, ET treatment reduces endogenous ABA levels, indicating that ET regulates heterophylly by suppressing ABA and regulating cell division and elongation [93] (Figure 1). 

GA concentrations in leaf primordia change in response to circumambient environmental cues, and the application of exogenous GA alters leaf complexity in different plant species. Furthermore, GA reduces leaf complexity by inducing leaf primordia differentiation and disabling the formation of marginal serrations and leaflets by suppressing transient organogenetic activity in the leaf margins [98]. The expression levels of the *KNOTTED1-like homeobox* (*KNOX1*) gene, a negative regulator of GA biosynthesis, were altered in response to submergence, and consequently, the accumulation of GA was changed in the leaf primordia. Variations in the expression of *KNOX1* appear to underlie differences in leaf shape [99,100]. Unsurprisingly, *t*GA has an adverse effect on heterophylly in aquatic plants [101]. Besides this, the effects of GA can either be enhanced by ET or inhibited by ABA. 

Auxin polarization is crucial for leaf primordia initiation and for the outgrowth of leaf lamina during leaf development [102,103]. Auxins are also involved in vascular patterning in leaves, which affects leaf morphology [68]. Therefore, auxins could play a role in heterophylly as downstream targets of other upstream phytohormones. 

In *R*. *palustris*, leaves acclimate by thinner epidermal cell walls and cuticles and by lying chloroplasts closer to the epidermis, which helps CO_2_ enter mesophyll cells through diffusion rather than stomata; an increase in the specific leaf’s area was also observed [104,105]. 

## 6. Formation of Adventitious Roots (AR) and Aerenchyma 

The formation of ARs is a typical adaptive response to flooding stress in many plants [60,87,106,107]. Submergence-tolerant species develop larger adventitious root systems than intolerant species, and these newly emerged adventitious roots often contain more aerenchyma [108]. ET triggers ARs growth, and other phytohormones are also involved in this process. ET can promote cell division in rice AR primordia [109]. Besides this, ET promotes adventitious root formation, either by increasing plant sensitivity to auxin or by stimulating the formation of root primordia in flooded plants [106]. 

In tomatoes, flooding induces ET synthesis and stimulates auxin accumulation and transport in the stem, which triggers additional ET synthesis and thus further stimulates a flux of auxin towards the flooded parts of the plant [19,60,110]. Although the endogenous auxin concentration remained unchanged in *Rumex* plants during waterlogging-induced adventitious rooting, continuous basipetal (shoot to the rooting zone) transport of auxin increased auxin sensitivity, which is required for adventitious root formation [58,59,106]. In addition, auxin activates plasma membrane H^+^-ATPases, leading to apoplast acidification [81] and the expression of expansin genes [82]. For example, the emergence of tomatoes’ adventitious roots is favored by cell wall loosening through the regulation of apoplastic pH or the upregulation of the expansin gene *LeEXP1*, which promotes cell wall disassembly and cell enlargement [111]. 

ABA might play negative roles in ARs’ formation during flooding stress. During partial submergence, the ABA content in AR primordia of S. *dulcamara* prominently decreased due to a downregulation of ABA biosynthesis and an up-regulation of ABA degradation [11,112]. The exogenous application of ABA negatively affects submergence-induced AR formation, whereas the removal of ABA using an ABA biosynthesis inhibitor induced AR emergence [58,59,60,112]. 

Furthermore, the role of JA in AR outgrowth seems to be ambiguous, and it depends on the plant species. Two JA-deficient *Arabidopsis* mutants produced more adventitious roots compared to the wild type, indicating that JAs function negatively in AR formation [113]. In contrast, Lischweski et al. found that JA acted positively on AR formation in petunias [114]. Cytokinin and JA are antagonists of auxin-induced adventitious root formation, but little information is available on how these regulators mediate adventitious root formation under submergence stress [114]. Transgenic Arabidopsis carrying the IPT (isopentenyl transferase, a key enzyme of cytokinin biosynthesis) gene exposed to waterlogging accumulated greater and faster quantities of cytokinins than WT plants. Cytokinin accumulation was accompanied by better chlorophyll retention and increased biomass and carbohydrate content relative to WT plants. IPT plants also showed an improved recovery ability [115]. The protective effect of CK was also reported in wheat. Transgenic wheat (*Triticum aestivum* L.) plants carrying the *ipt* gene were more tolerant to flooding than wildtype plants, with a higher yield and less growth inhibition during flooding [116]. Cytokinins might play a role due to their nature of activity (cell division promotion and cell expansion, but also senescence delay) not only in the roots, but in the above ground plant organs as well, probably even in the escape strategy.

Under waterlogging conditions, the content of SA was significantly increased in waterlogging-tolerant soybean cultivars, which may further stimulate ARs and enhance gas exchange and finally show tolerance to waterlogging stress [117]. A proper concentration of SA significantly increases the adventitious root formation in soybeans, and it increases SA-triggered PCD responses and peroxidation, and therefore, aerenchyma cells develop in the root, which improve oxygen supply. Secondly, accumulated SA may stimulate adventitious root primordium formation [118] (Figure 1). 

ET is implicated in the development of aerenchyma, which is the intercellular space that facilitates gaseous exchange and maintains the physical strength of tissues. The development of secondary aerenchyma in a flooded soybean was inhibited in response to the exogenous application of ABA, indicating that aerenchyma formation requires a reduction in ABA content [119,120,121,122,123]. ABA promotes suberin deposition in cell walls to regulate abiotic stress responses. For soybean aerenchyma cells’ development, a root cell must be unsuberized; therefore, the biosynthesis of suberin must be suppressed by the down-regulation of ABA. In waterlogging-tolerant plants, aerenchyma cells were well-developed, with a significant decrease in whole plant ABA contents compared to sensitive and control plants [124,125,126]. 

Besides aerenchyma, hypertrophied lenticels are formed to facilitate the connection between underwater hypoxic tissues and the atmosphere [123,127]. In *Glycine max*, hypertrophied lenticels were the swollen tissues at the stem base, which was attribute to radial cell division and expansion [123]. Hypertrophied lenticels are associated with auxin and ET production and are observed in many gymnosperms and angiosperms during flooding [123,128]. 

## 7. Unanswered Questions and an Outlook 

Most studies have paid close attention to the detrimental effects of a low-oxygen environment on plant metabolism, whereas the molecular responses and signaling events occurring in the reoxygenation stage have been largely ignored. However, when floodwater subsides, high levels of ROS (reactive oxygen species)-related lipid peroxidation are observed in plants during reoxygenation [129]. Although excessive water exists in the soil, some plants show symptoms of water deficiency after reoxygenation, as evidenced by wilted leaves and the induced expression of dehydration-responsive genes. Thus, the post-flooding period is associated with oxidative and drought-induced damages, and a true resilience to flooding should include the ability to tolerate both post-flooding and flooding phases [129,130,131,132]. Enhanced expression of ET biosynthetic enzymes was found in *Arabidopsis* during reoxygenation, and the ET-insensitive mutants ein2-5 and ein3eil1 exhibited impaired genes expression associated with ABA biosynthesis, dehydration, and enhanced sensitivity to post-anoxic stress [132]. The role of ABA in regulating post-submergence responses, its interaction with ROS, and its control of plant ionic homeostasis during post-submergence recovery are some interesting aspects for future research. 

In Arabidopsis, the transcript levels of JA biosynthesis genes increased, and jasmonates accumulated rapidly during reoxygenation, and the JA-inducible accumulation of antioxidants may relieve oxidative damage caused by reoxygenation and improve plant survival after submergence [132,133,134]. Although ABA has always been regarded as a negative regulator of the development of morphological adaptations to soil flooding in different plant species, recent evidence suggests a positive role of ABA in the modulation of plant responses to hypoxia and the ability to recover during the post-hypoxic period [73]. In rice, the flooding tolerance-associated SUB1A gene confers tolerance to drought and oxidative stress during reoxygenation through increased ROS scavenging and enhanced ABA responsiveness [129]. 

Targeted protein proteolysis via the N-end rule pathway is evolutionarily conserved, and it is a key mechanism involved in the low-oxygen response in plants, and ERFVII transcription factors are substrates for the N-end rule pathway, acting as oxygen sensors [135,136]. In recent years, NO has arisen as an important modulator of low oxygen conditions [137]. Under hypoxic conditions, NO depletion is regulated by ET through the RAP2.3 ERFVII transcription factor in Arabidopsis [138], which, in turn, promotes the stabilization of ERFVII transcription factors [139]. NO has been shown to act downstream on ABA signaling in the regulation of seed germination and stomatal conductance (at least partially [140]). Moreover, it has been shown that NO might antagonize ABA signaling through posttranslational Tyr nitration of the PYR/PYL/RCAR ABA receptors that significantly reduce their PP2CA inhibitory activity in the presence of ABA [141]. Gibbs et al. [142] reported that NO and oxygen in Arabidopsis seeds promote the degradation of ERFVII, leading to the downregulation of ABI5 and germination. Conversely, recent evidence supports the involvement of ABA, at least partially, in the induction of NO biosynthesis genes’ expression in hypoxic conditions [143], suggesting a complex crosstalk interaction between NO, ABA, and ET. Thus, it is plausible that that ET can interact with NO (nitric oxide)–oxygen sensing pathways in concert with ABA and modulate plant physiological and metabolic responses to low oxygen environments. Specific details of this interaction warrant a separate investigation. 

## Figures and Tables

**Figure 1 plants-10-01982-f001:**
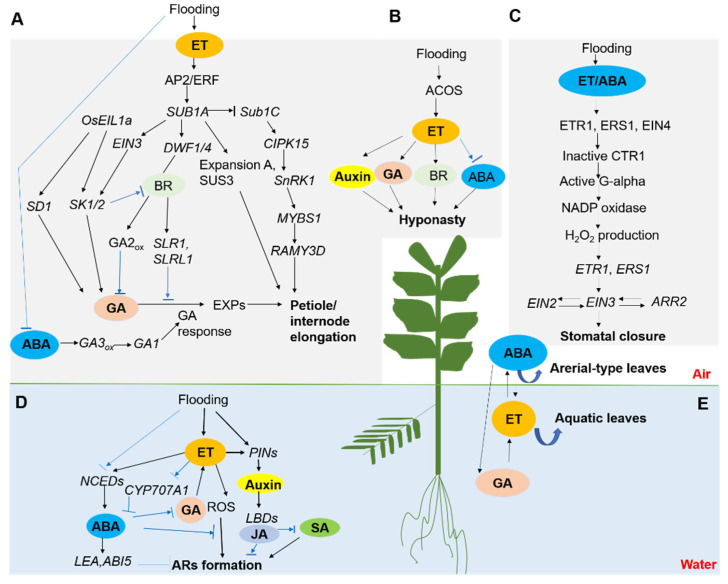
A schematic model of plants’ morphological adaptations during flooding stress (adapted from [17,33,68,69,70,71,72,73,74]). To cope with flooding stress, plants undergo multifaceted anatomical, metabolic, and morphologic alterations. The morphological adaptations include shoot elongation (**A**), hyponasty (**B**), stomatal closure (**C**), adventitious roots formation (**D**), and heterophylly induction (**E**). The model is collective and integrates findings reported in one or more species. The interactions and hierarchy of signaling components can vary depending on species. (**A**): Processes, hormones, and genes involved in submergence-induced shoot elongation (blue arrows indicate an inhibitory effect). Submergence causes accumulation of ethylene, and subsequently, it induces reduction of ABA biosynthesis and promotes the ABA catabolism, which leads to a lower endogenous ABA concentration in rice. This stimulates GA signaling and ultimately enhances petiole elongation. BR is also involved in this process. (**B**): Increased ethylene biosynthesis in waterlogged Arabidopsis is linked to transcript accumulation of ACO5. GA, BR, auxin, and ABA positively regulate the hyponasty, while the accumulation of ABA was inhibited by ethylene. (**C**): A proposed model for the hormone-mediated stomatal closure under waterlogging stress. Ethylene and/or ABA directly or indirectly enhance H_2_O_2_ production under waterlogging conditions. The binding of ethylene to ETR1, ERS1, and EIN4 induced the inactivation of CTR1, resulting in the activation of G alpha, which promotes H_2_O_2_ production via NADPH oxidases. ETR1 and ERS1 translocate the signals of H_2_O_2_ to EIN2, EIN3, and ARR2, which are essential for stomatal closure functioning. (**D**): Adventitious root biogenesis and growth regulation in plants. Ethylene, which accumulates in submerged tissues, promotes adventitious root growth. GA enhances ethylene-induced root growth, while ABA acts as a root growth inhibitor. ROS act downstream of ethylene to mediate the root growth response. Auxin also promotes adventitious root biogenesis. (**E**): In aquatic plants, ABA could initiate and maintain the development of aerial-type leaves, and this process is dependent on the cross-talk with ethylene- and GA-signaling pathways, which initiate and maintain the formation of submerse leaves. Abbreviations—AP2/ERF: apelata2/ethylene response factor; CYP707A1: encode abscisic acid 8′-hydroxylases; ROS: reactive oxygen species; PINs: pin-formed protein; LBDs: lateral organ boundaries domain; ARs: adventitious root; DWF1/4: dwarf1/4; BR: brassinosteroids; SD1: semidwarf1; SK1/2: snorkel 1/2; EIL1a: ethylene insensitive like 1a; SLR1: Slender Rice-1; SUS3: sucrose synthetase 3; SnRK1: SNF1-related kinase 1; CIPK15: calcineurin B-like–interacting protein kinase gene; ACO: 1-aminocyclopropane-1-carboxylic acid oxidase, ACC oxidase; ACS: ACC synthase; AMY: amylase; EXP: expansins; SUB1: submergence1; NCED: 9-cis-epoxycarotenoid dioxygenase; GA3ox: gibberellin 3-oxidase; ARR2: 2-component response regulator; CTR1: constitutive triple response 1; EIN: ethylene insensitive; ERS1: Ethylene response sensor 1; ETR1: Ethylene receptor 1; G alpha: G protein alpha subunit; ARR2: 2-component response regulator.

## Data Availability

Not applicable.

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
