# Peer review of "Understanding a Mechanistic Basis of ABA Involvement in Plant Adaptation to Soil Flooding: The Current Standing"

_plants, 2021, doi:10.3390/plants10101982_

Round 1
Reviewer 1 Report
The manuscript reviews the role of ABA in plant processes under flooding conditions. The topic is interesting and reviewing contradicting results in research is very useful deed. The manuscript generally reads well, however, English editing is a must – there are sometimes wrong tenses used, there are often missing or incorrect articles etc. Some abbreviations – GA, ACC, are not explained. In the reviews, the tables and figures summarizing knowledge on a given subject are of great value. In this manuscript, the tables need a little bit more detail to fulfil their capacity. Also, my main impression is that at the beginning of the manuscript preparation, the authors had time and were very conscientious while writing; but as the deadline was nearing, the authors started to rush and this hurry is reflected in the writing quality. There are sometimes bits of information piled without any linkage. The last chapter (7. Unanswered questions and an outlook) is a bit chaotic and quite unpleasant to read. The last paragraph summarising what should be done in future is not very specific and deserves, yet again, more detail.
Overall, it would be a great shame not to realise the full potential of the manuscript.
Table 1: Pisum, not Plsum
Table 2: Specify better the column chemicals/mutants – which gene was knocked out/overexpressed / exogenous application? What concentration and timing for exogenous application? This is too vague, there is much greater potential in the informative value of the table. Also, there is a capital letter in Glycine and correct Oryza (not Oryz).
Lines 70-73 – Clarify in which way it is an antagonist to GA?
120-124 – do all species that respond to flooding by ABA decrease really employ escape strategy? It is an interesting theory and it deserves a little bit more detail.
330-332 – This is the only time you mention cytokinins, even though they might play a role due to their nature of activity (cell division promotion, cell expansion, but also senescence delay) not only in the roots, but in the above ground plant organs as well, probably even in the escape strategy. Is there any literature concerning these issues?
357-367 – rewrite this paragraph please. The circadian clock influences almost everything in the plant, that is not surprising, and it is not clear, what you want to say.
359 – 360 – reference, please.
361-363 – too vague, explain in more detail, please.

Reviewer 2 Report
This paper deals with Abscisic Acid as a Regulator of Plant Adaptive Responses to Soil Flooding. The paper needs serious improvements before considering it for publication. Please see below my suggestions:
According to the Plants Instructions for authors - please see the link https://www.mdpi.com/journal/plants/instructions Acronyms/Abbreviations/Initialisms should be defined the first time they appear in each of three sections: the abstract; the main text; the first figure or table. When defined for the first time, the acronym/abbreviation/initialism should be added in parentheses after the written-out form. Please check and revise all abbreviations used in the manuscript, beginning with the Abstract, L17, ABA - explain it in full before abbreviating it; L70 GA; etc.
Keywords must be separated by semicolon.
L65-67. Please highlight better the aim of the study or the special aspects that this research brings to the topic, underlining also the novelty character of it. Maybe you can also justify why you chose this topic.
Tables 1 and 2, last column. Please delete the name of the authors. It is enough the inserted reference number.
Please check once more the Instructions for authors regarding the size of the characters that must be used in the manuscript, titles of the tables and figures, content of the tables - and revise them accordingly.
I suggest a new section/subsection Fertilizers and climate changes influences - or similar (maybe of the section 2 where "Changes in ABA content and ABA- regulated responses during submergence" are described) because in the manuscript it is about the ABA in the case of several plants, many of them being cultivated. Most of the times, the respective crops are subjected to the action of fertilizers, pollutants and climate changes / effects. Therefore, at the time of floods, these plants (and therefore also ABA) are subjected to "modified" (attenuated or diminished) actions exerted by soil substances. Moreover, the climate also shows its major influence, a dry / warm climate having obviously different effects from a humid / cold one. Please check and refer to: Bungau et al. Expatiating the impact of anthropogenic aspects and climatic factors on long term soil monitoring and management. Environ Sci. Pollut. Res. 2021, 202, 30528-30550. https://doi.org/10.1007/s11356-021-14127-7 ; Samuel et al. Effects of long term application of organic and mineral fertilizers on soil enzymes, Rev. Chim., 69(10), 2018, 2608-2612. https://doi.org/10.37358/RC.18.10.6590 and Samuel et al. Enzymatic indicators of soil quality. J. Environ. Prot. Ecol. 2017, 18(3), 2017, 871-878. Gitea, M.A. et al. Orchard management under the effects of climate change: implications for apple, plum, and almond growing. Environ Sci Pollut Res. 2019, 26, 9908–9915. https://doi.org/10.1007/s11356-019-04214-1
For this new section/subsection I suggested, you can make also a scheme or a table, describing all the factors mentioned in the literature influencing the content/concentration of ABA in plants, of course with a last column of Ref.
References. Please complete all the data requested for references according to the instructions for authors: Journal Articles:
1. Author 1, A.B.; Author 2, C.D. Title of the article. Abbreviated Journal Name Year, Volume, page range.
Round 2
Reviewer 2 Report
I checked the manuscript. The authors responded to my requests, highlighting in red all changes.
Author Response
Thanks for effort to improve our manuscript.